# Exquisite Energy Savings at Cold Metal Forming of Threads through the Application of Polymers

**DOI:** 10.3390/polym14061084

**Published:** 2022-03-08

**Authors:** Miroslav Píška, Petra Sliwková, Zuzana Vnuková, Martin Petrenec, Eva Sedláková-Valášková

**Affiliations:** Faculty of Mechanical Engineering, Brno University of Technology, Technická 2896/2, 616 69 Brno, Czech Republic; piska@fme.vutbr.cz (M.P.); 182656@vutbr.cz (Z.V.); martin.petrenec@mubea.com (M.P.); eva.sedlakova@mubea.com (E.S.-V.)

**Keywords:** energy, polymers, cold forming, thread, wear

## Abstract

One of the global problems today is energy—its production and distribution. As the human population grows, the consumption of energy rises simultaneously. However, the natural sources are limited, and so the focus on power savings becomes more and more important. One of the ways to reduce consumption is the use of effective lubricants and tribological fluids in industry, especially in processes with high demands on energy but high quality of products as well. Forming is a typical example of such technology, and the application of polymers seems to be a very important challenge, because the application of straight oils or lubricant with extreme pressure additives seems to be prevailing in that field. Nevertheless, the polymer lubricant should fulfill all European standards as well as the environmental and ecological limitations with respect to health and the natural environment and its recycling and disposal. This paper is focused on the forming technology of threads and the application of selected polymers to the forming process. The measured and quantified criteria are torque and force loadings, energy consumption, and quality of the produced surfaces. Kistler dynamometers, scanning electron microscopy, and advanced surface topography with the use of Alicona IF-G5 were applied to assess all aspects of the tribological and energy aspects of six modern process fluids, three lubricating pastes, and two fluid modifications. The results show that the polymer synthetic lubricant (at volume concentration 20% in water) can reduce the total energy consumption by up to 40% per forming cycle (in mean values) at average surface roughness below 0.8 μm.

## 1. Introduction

It is not easy to specify all characteristics of process fluids. Normally, they can be defined by physical and chemical properties, but the technological effects are crucial as well as their health aspects and environmental risks. Testing of mechanical properties is mostly standardized, like many tribological tests. However, they hardly reflect the real loading and behavior of the tools in manufacturing practice. Interpretation or a correlation can be difficult or even misleading, because the reality is more demanding and severe conditions are normally used. Some applications also invoke a use of very powerful and rigid machines, and it is difficult to install some sensors in such conditions and places. A typical problem is forming technology, especially drawing of wires, stamping, or forging, and the price of the tools bought just for a testing with an acceptable statistical significance is very expensive.

A chance to substitute the real and costly technological research with a similar but cheaper technology can be, for example, form tapping. This technology is studied widely in works [1,2], with high impact on the precision and integrity of the tapped surfaces. A comparison between the cut and formed thread was also made in [3], with a special impact on PVD coatings of the tools. The beneficial effect of diamond coating was studied in [4] for threading of aluminum-based alloy that tends to make so called built-up edges, affecting many parameters of the threads. However, the effect of the cooling or lubricating fluid from many aspects was studied in works [5] and also in [6], with high consideration for the natural environment.

A new approach for selection of the polymer as the process fluid can be seen in [7], where polymer–water-based cutting fluid can effectively substitute mineral oil-based fluids in grinding processes. The discovery of fullerene and nano-additives started a new epoch of application of unique materials [8,9] at extreme conditions of loading. The nanoparticles in the forming process were studied by the authors in [10], with a tremendous effect on the forming process of metal containers. All of these works and many others [11,12,13,14,15,16,17,18,19,20,21] inspired the authors of the paper to thoroughly study the effect of polymers in a forming technology from several aspects. The selection resulted in the production of thread by forming taps.

The tools—forming taps—are much cheaper in comparison to real forming tools; the exam can be repeated and statistically assessed. There is a similar forming process with high contact loading where the process fluid presents a significant role with impact on the quality of the produced surfaces. Some other specific variables can also be derived from the test.

The unique contribution of the paper is that it integrates the deformation work, coefficient of friction, specific forming force, and surface morphology for a wide range of polymers in the forming technology of threads.

## 2. Theory

Thread forming is performed using a fluteless tap, which closely resembles a cutting tap without the flutes [22]. There are lobes periodically spaced around the forming tap that actually perform the thread forming as the tap is advanced into a properly sized hole. Since the tap does not produce chips, there is no need to periodically back out the tap to clear away chips, which, in a cutting tap, can jam and break the tap. So, the forming of threads starts with the operation of pre-drilling the hole to a diameter near to the mean diameter of the thread. Next, the hole is chamfered to make for better entry of the tap tool. Finally, a sequential forming of the individual cross-sections of material by the tap follows—Figure 1. This process is accompanied by very intensive friction and the plastic flow of workpiece material. The use of a proper lubricant is very essential in this technology. However, the hardening of the deformed layers promotes the material’s resistance to fatigue, corrosion, etc. An inner metal box placed in a machine allows the use of a wide series of fluids in limited volume without any contamination of the machine (pipe, pumps) by the different products.

## 3. Materials 

### 3.1. Polymer Characterization

Eleven different polymer materials (a soap, fluids, and pastes) in the process of thread forming are compared in this study. Four of them are used in full concentration; seven of them can be diluted (with water). Each of them is recommended to be used as a suitable process fluid for cold tapping and forming processes by their mass-producers, with effective impact on friction, tool wear, quality of production, etc. A brief and very limited description (due to respect to company know-how and no intended advertisement of the products and brand names) follows:***1.*** ***The polymer soap “A”***

A dry lubricant based on calcium Ca(OH)_2_ and sodium soaps (NaOH), potassium hydroxide (KOH), and other ingredients, made in granular shapes—Figure 2—with a characteristic yellowish color and soap smell. A well-known effect on shearing forces and friction [23] has been confirmed by many similar industrial applications and also by the author’s team experience. If combusted small parts such as aldehyde, ketone, and ester get into the air, these things could cause stimulation of the mucous membrane, in the worst-case causing cancer. The melting point is around 120 °C; the combusted point is over 200 °C. Powder density is 700 ± 50 kg/m^3^. The polymer powder has good water solubility at room temperature (around 20 °C); the pH value for polymer powder was 12 for a temperature of 90 °C at a concentration of 10 g/L in water.

***2.*** 
**
*Polymer Paste “B”*
**


This is a polymer paste suitable for heavy drawing operations, especially for carbon wires. It is a high-viscosity compound that is intended for a use in reservoirs during forming. The producer presents the product as chlorine-free, brown in color, and with a low environmental impact. It is formulated as a natural substitute for sodium and calcium powders and graphite and molybdenum disulfide powders used for forming. This product is without any inauspicious harmful physical or chemical impacts on life and the environment. The burning point is greater than 180 °C (ASTM D 93-18) [24], the relative density 0.930 kg/dm^3^ (ASTM D 1298-12b) [25], and viscosity (40 °C) 216 cSt (ASTM D 445-21e1) [26].

***3.*** 
**
*Polymer Paste “C”*
**


This polymer paste, which was tested, is chlorine-free and a suitable lubricant for the forming process. Throughout the forming process, it should be stored in tanks, because it is a high-viscosity compound. It is a polymer paste suitable for heavy drawing operations, particularly for carbon steel wires. This product is without any inauspicious physical or chemical impacts on life and the environment. The burning point is greater than 190 °C (ASTM D 93-18) [24], the relative density 0.960 kg/dm^3^ (ASTM D 1298-12b) [25], and viscosity 216 cSt at 40 °C (ASTM D 445-21e1) [26].

***4.*** 
**
*Polymer Fluid “D”*
**


A synthetic highly viscous brown polymer fluid suitable for heavy drawing operations of hot-rolled material which is chlorine-free. The special film at the drawing material is without the deposition of phosphate film, with low environmental impact. This polymer was specially formulated to become suitable for chlorinated compounds at material drawing. The producer offers excellent surface adhesion properties, anti-wear characteristic, and high resistance to extreme pressure. This polymer is also suitable for high-alloy carbon steels. The specific gravity at 15 °C was 0.978 kg/dm³ (ASTM D 1298-12b) [25], viscosity at 40 °C 1.095 mm^2^/s (ASTM D 445) [26], and flash point (PM) 93 °C (ASTM D220) [27]. This product was tested at 100% concentration only (as available on the market).

***5.*** 
**
*Polymer Fluid “E”*
**


A water-soluble polymer cooling fluid (Tech Cool 35632 BF, hebro chemie, BASF, Germany) for the most demanding sulfur-bonded machining; fully transparent polymer liquid, free of mineral oil, boric acid, and formaldehyde. The liquid does not emulsify foreign oils and has excellent cooling, detergent, and cutting properties and excellent corrosion protection. The solution is prepared by diluting the concentrate with water, with a refractometric liquid factor of 1.4%/°Brix. According to the safety data sheet, the liquid causes serious problems and is harmful to aquatic organisms, with long-term effects. This product was tested at several concentrations according to the producer´s recommendation.

***6.*** 
**
*Polymer Fluid “F”*
**


Yellowish polymer fluid for wet drawing and wire forming with a characteristic smell. The dangerous substances in the polymer with concentrations less than 5%: 2-amino-2methylpropanol; and less than 0.5%: poly[(dimethylimino)ethylene(dimethylimino)ethylenoxyethylen]. For manipulation with the polymer, it is needed to have at least gloves made from nitril resin elastic with in-use exposures greater than 30 min and thickness greater than 0.4 mm. Acidity of the polymer: pH value (100 g/L and 20 °C) 8.1; viscosity of 270 mm^2^/s at 20 °C; and boiling point higher than 100 °C (the product is without determination to spontaneous combustion). The specific density was 1.05 g/cm^3^. It has a good water solubility in a wide range of concentrations. In the normal conditions, it is stable, without any decomposition products. It is free of boron, formaldehyde depots, monoethanolamine, and other critical ingredients. This product was tested at several concentrations in this study.

***7.*** 
**
*Polymer fluid “G”*
**


Polymer fluid is cooling fluid based on 4% of sulfur and two other compounds, without any chlorine. The carcinogenic influence of the polymer fluid in people is less than 3%. There is a need to have special conditions to guarantee an effective deaeration in the hall to avoid a fire due to production of an aerosol. The second important condition is to meet the common hygienic rules during handling of the fluid and to wash hands regularly at work. The color is mostly yellow, like flower honey, due to natural ingredients. The pH value is 9.2 (50 g/L, 20 °C). The melting point, first watering point, combustibility, evaporation rate, and steam pressure were not noticed. The burning point is higher than 130 °C (ISO 2592). The density is 0.99 g/cm^3^ at 20 °C (DIN 51757) [28]. Kinematic viscosity is 66 mm^2^/s (40 °C) according to the DIN EN 16896 [29]. The chemical stability was very good without any dangerous reactions, but the product should be without any contact with high temperatures, hot spots, sparks, open fire, and other ignition options. It is not classified as an acute toxicity material. This product was tested at several concentrations in this study.

***8.*** 
**
*Polymer fluid “G+”*
**


In fact, this was an experimental trial: the polymer fluid “G” enriched by an extra 1% of EP additives. This mixture was tested at several concentrations in this study. This mixture is not commercially available.

***9.*** 
**
*Polymer fluid “H” with nanoparticles WS_2_*
**


A metallurgical water solubility emulsion fluid, basically for the cold forming process on the base of very strong disulfide wolfram (WS_2_). The properties of the fluid include prevention against extreme pressure and high efficiency against abrasion and friction. These properties help increase the durability and tool life. This improves the quality of the surface, thus reducing the work necessary before welding. This makes milling under extreme pressures without smoke possible, thus saving time and money. The seller recommends a concentration around 2–10% with water, but it should be adjusted in each particular application. The technology of this fluid is based on the multilayer of the nano-fullerene particles with a formed disulfide wolfram (WS2). These multilayer IF-WS_2_ nano-bullets are well known for their good temperature stability from −273 °C to +500 °C at pressures of 4,263,000 PSI. It can be applied universally in extreme conditions, such as high pressure and low temperatures, high pressures and vacuum, high loads, and high torques with corrosion resistance. The sizes of the nanoparticles range from 50 to 200 nm, which easily fill all kinds of “indirectness”, such as microcracking and scratches. During the high “charges” pressures (about 1 GPa), the layers separate from the “bullets” and obtain a thin protective layer at the surface steel, which decreases the friction and degradation between material and tool. In the “substance” polymer fluid, there are some special hazardous combustion products (carbon monoxide, carbon dioxide and sulfur oxides). The nanoparticles have the characteristic dark gray color with a mild ammoniacal odor. The density is 9.09 lb/gal at 15.6 °C. Polymer fluid “H” with nanoparticles is a nontoxic material, without active sulfur, chlorine, boron, formaldehyde, and zinc and without critical amin (such as monoethanol or dicyclohexylamines). This material is not classified as carcinogenic or germ cell mutagenic. This product was tested at several concentrations in this study.

***10.*** 
**
*Polymer fluid “I” with nanoparticles WS_2_ and MoS_2_*
**


In general, a suspension of a heterogeneous mixture of WS_2_ and MoS_2_ inorganic fullerenes in the form of complex nano-sheets of sizes from 50 to 200 nm, made of metal atoms sandwiched between two sublayers of sulfur. The solute particles do not dissolve but become suspended throughout the bulk of the solvent. “A strong tendency to sedimentation of the nanoparticles”; the nanoparticles tend to settle at the bottom of their container. The properties include excellent resistance to high pressures, high load, and high torque and good corrosion resistance. Nontoxic material without chlorine, boron, formaldehyde, zinc, and critical amin (such as monoethanol or dicyclohexylamines). The nanoparticles have a characteristic dark gray color and a characteristic odor. The density is 1.185 g/cm³ at 25 °C. This material shall not be classified as carcinogenic or germ cell mutagenic. This product was tested at several concentrations in this study.

***11.*** 
**
*Polymer fluid “G++”*
**


In fact, this was also an experimental trial: the polymer fluid “G” enriched with polymer fluid “F”, containing the nanoparticles WS_2_. This product was tested at several concentrations in this study, and the mixtures are not commercially available.

### 3.2. Tested Material and Tooling

The tested material was low-alloy chromium steel 54SiCrV6 in a hot-rolled state, and after the thermal hardening it was formed into blocks—Figure 3. The chemical and mechanical compositions according to CSN EN 10089-03 [30] are listed below in Table 1 and Table 2, and the analysis of water used for diluting the polymers can be found in Table 3.

The threading technology consisted of drilling of the pilot hole with solid carbide drills ø5.8 mm, gripped with thermogrip Bilz–HSK A63 (v_c_ = 70 m/min, f = 0.10 mm), and chamfering of the corners with tapered solid carbide counterborer 90° (v_c_ = 50 m/min, f = 0.05 mm). Then, a blinding of the hole with a gummy stopper (to preserve the polymer leakage from the hole) followed, and, finally, the thread was formed with the HSS-E cold forming tap M10-6HX InnoForm1, PVD-coated (monolayer of TiN, thickness of coatings: 3.8 ± 0.18 μm, Ra = 0.8–0.9 μm)—Figure 4. The forming taps were positioned in the compensation adapter Emuge Franken KSN Synchro IKZ (Emuge Franken, Nürnberger, Germany) for the push-pull loading. Rotational speed and feed per rotation were kept the same for all tests (number of rotations *n* = 3000 min^−1^, f = 1 mm). Before measurement, all taps were cleaned and controlled for each polymer, and new forming taps were used for each concentration and sort of fluid. The quality of the tools and the threads was measured and analyzed with the flexible optical 3D measurement system Alicona-G5 InfiniteFocusG5 (Graz, Austria) and with scanning electron microscope LYRA or MIRA3 (TESCAN, Brno, Czech Republic).

### 3.3. Experimental Method

Threading tests were performed by 5-axis machining center MCV 1210 TAJMAC-ZPS, (Zlin, Czech Republic), with control system SINUMERIK 840D (SIEMENS, Munich, Germany). Each hole was blinded at the bottom with a little gummy stopper and pre-filled with the tested polymer. Each tool was submerged into the polymer before its use. Strong liquid flow and flooded conditions were applied when forming the threads. The exact application of process lubricant differed according to the sort of polymer in order to make the lubrication inlet as good as possible—Figure 5. The torque moment and the axial force were measured by the piezo-electric dynamometer Kistler 9272 and charge amplifiers 5011. The DynoWare software for universal data acquisition, with dynamometers and Kistler charge amplifiers, Type 2825A, (Kistler, Winterthur, Switzerland) at sampling frequency 1000 Hz per each variable, was used. The volume concentrations were prepared with use of standard laboratory graduated cylinders and syringes.

The mathematical analysis of the cross-section of the material to be formed into thread confirmed a polynomial curve of the fifth order according to its length—Figure 6.

The whole process of thread forming, with highlighting of some critical time periods, can be seen in Figure 7.

The period I reflects the production of the first thread profile. Moreover, it helps to analyze the design of the forming tool as well and the equality of the loading.

The period II is an important part, because it should reflect a stabilized thread production, and, in that case, the specific forming force important for formability of the material can be derived. If there is a rise, it reflects material hardening, a poor lubrication in the forming process, or tool wear. If there is a fall, it may reflect a wear or even breakage of the tool. It can be expressed in the relation of the mean torque moment *M*_t_, deformed cross section *A*_D_, and radius of gravity of the active force *r*_s_:
(1)$$K_{f}=\frac{M_{t}\times 1000}{A_{D}\times r_{s}}$$

The following period III is typical for a run-out of the tap. The thread is already finished, but the tool stays squeezed by elastic forces by the surroundings in close vicinity. This time period can be used for estimation of the coefficient of friction as a ratio of tangential force *F*_T_ and normal force *F*_N_ (in general), which can be derived from the torque moment *M*_T_, axial force *F*_f_, and radius of gravity of the active force *r*_s_:(2)$$\mu =\frac{F_{T}}{F_{N}}=\frac{F_{f}\times r_{s}}{M_{T}\times 1000}$$

The last period IV gives information about passive forces in the material after forming. It is a sort of spring back effect, which can point to the effectiveness of technology or tool wear intensity.

A typical time series of the torque moment and axial thrust force can be seen in Figure 8. The individual periods were statistically assessed (normal distributions, mean values, standard deviations) with Excel, Windows 10.

From the time series of torque *M*_T_, axial force *F*_f_, and forming conditions (rotational speed *n*, feed per rotation *f*, and time increments Δ*ti*), the whole deformation work can be numerically calculated and integrated as a sum of the work *A* done by the torque *A*_MT_ and translation force *A*_Ff_
(3)$$A=A_{\mathrm{MT}}+A_{\mathrm{Ff}}\;$$
(4)$$A_{\mathrm{Mt}}=\sum \frac{M_{\mathrm{Ti}}\times n\times \Delta ti}{9.55}$$
(5)$$A_{\mathrm{Ff}}=\sum \frac{F_{\mathrm{fi}}\times f\times n\times \Delta ti}{60,000}$$

After the forming, a basic measurement of the thread was performed with use of thread gauge M6 × 1-6H (Meusburger, Wolfurt, Austria). Furthermore, the samples were cut longitudinally in the middle—Figure 9—and measured by Alicona IF G-5 and analyzed with the electron microscope TESCAN Vega working in the regime of secondary electrons (SEM).

The cross-sections of samples were cut by the diamond saw Struers Discotom 100, then ground with water cooling and use of sandpapers of grain grits 80, 120, 280, 400, 600, and 1000 (according to FEPA-F). Then, polishing with diamond water suspensions (Struers Dia-Pro of 2, 1, and 0.5 μm) under ethanol on satin followed with the machine Saphir 250M1 to obtain a glossy surface appearance. Subsequently, some specimens were etched with Nital (nitric acid in alcohol, volumetric concentration 2%).

## 4. Results and Discussion

All tested samples and their variables were characterized graphically by their mean values and standard deviations in Table 4, Table 5, Table 6 and Table 7 and in Figure 10 and Figure 11 (and in Appendix A in the Appendix A). The values represent experiments repeated at least five times, always with a new tool, for one concentration, so that several thousands of data points in total were assessed with statistics for Gaussian normal distribution.

### 4.1. Evaluation of the Deformation Work, Coefficient of Friction, and the Specific Forming Force

Generally, the diagram shows that the full concentration of the product does not mean the best results in the studied parameters. On the contrary, a dilution of the concentrate enhances the tribological effect, especially lubrication and friction. It affects the shearing forces and energy consumption. Coefficient of friction does not reflect the forming process, because it is a specific variable, a ratio that does not express the total magnitudes of the components. The specific forming force is a very useful variable, because it reflects the formability of the material at specific conditions and affects the wear and its intensity as well.

Excellent results were obtained for the use of the soap “A” in granulated form. This confirms the daily experience in many companies. If the tool surfaces and deformed material can firmly catch the particle, then the low shearing forces at the interface are guaranteed—Figure 10. However, there is the problem of the dispersion of the data, and so the benefit depends on the success in whether the particle can be captured by the surface irregularities and then effectively sheared. However, the particles are not uniform, their grain sizes vary—Figure 2 and Figure 5a. Possibly a similar problem might occur with the product “B–D”, which could spin off from the tool. On the other hand, the results support high applicability of the polymers in general, diluted to 10–20%—Figure 11 (and Appendix A in the Appendix A to the paper). In other words, these materials are much more effective in the surface penetrations and act at the interface effectively. The wetting ability enhances the protective effect and good performance of the liquid polymer “E” as well as the expected sulfur-bonded forming—Figure 11. Such excellent results have not been confirmed for the fluids marked “F” and “G” unfortunately.

The beneficial effect of the nanoparticles WS_2_ in the polymer “H” was also proved, because the reduction of the deformation work was simply amazing. Possibly, during the forming operation, the nano-flakes could self-arrange in the formation of parallel lubricating layers helping to achieve a better plastic flow of material, thus reducing friction and mechanical work. Unfortunately, a complex or a synergetic effect of WS_2_ and MoS_2_ particles was not proved at the polymer “I”, and no remarkable effect was shown for the fluids “G+” and “G++” either. In other words, a research of the nanoparticle bondings to the polymers and their anchoring to the tool/steel interface should be studied more.

### 4.2. Evaluation of the Formed Thread Surface Quality

In fact, the thread profiles are not geometrically as perfect as those made by cutting, however, because the crests do not cover the whole cross-section thread area—Figure 9.

The standard surface parameters such as average roughness also cannot be regularly evaluated, because it does not meet the fundamental conditions according to the CSN EN ISO 4287 standard. However, approximate assessment can be performed by means of optical microscopes (Alicona IF G-5) and scanning electron microscopy (Tescan Vega or Mira-3). On the basis of many observed surfaces made by the forming in this study, for most of the surfaces, four categories of surface have been found—Figure 12 and Figure 13. Low values of deformation works, coefficients of friction, and specific forming forces support formations of smooth surfaces. It is also reflected by low dispersion of data, their variances, and standard deviations. Nevertheless, the question of production of the best surfaces is still open, because we did not significantly evaluate the passive forces that contribute to the formation of surface layers.

### 4.3. Wear of the Forming Taps

This study was not intended to measure a tool life of the taps, but, due to very fine martensitic material structure and very low plasticity of the material (the yield strength and tensile strength are very close), wear was frequently observed. In general, if the coefficient of friction reached the value above 0.5 or the specific forming force began to vary, the wear of the tool was confirmed—Figure 14. The basic mechanism of the wear was abrasive, and after losing the protective TiN coating, it turned to catastrophic intensity or breakage. The main difficulties were observed with product I, which also exhibited (apart from the squeaky sounds when forming) several fractures, and so the product of the technology used in this way is more than disputable. A modern method which can be used to analyze the volumetric intensity can be seen in Figure 15, but it is so far very labor-intensive and arduous.

The application of polymer soaps is still valid and effective, but there is a poor guarantee for a uniform spreading of the soap around the drawn wire—see Figure 16. Moreover, if the material undergoes a heat treatment, the danger of cancer is not acceptable by many countries around the world today. Another problem is a fast sedimentation of the nano-products that can initiate many other problems with mixing of the fluids, machine maintenance, dermatological problems, etc.

## 5. Conclusions

This research was focused on evaluation of the polymers at the forming process. It confirmed a very good applicability and acceptable pricing due to relatively low concentrations supporting the tribology and plastic deformation of the tested steel. The right selection of polymer, a suitable dilution, a pressured application, and appropriate maintenance can be the reliable path to future forming technology and new standards. The recommendation of polymer E seems to be very appropriate, as well as the application of the nano-additives. However, to prevent sedimentation, a mechanism similar to polarization of oil particles via emulsifiers (keeping them dispersed in the fluid) should be found. The supportive recommendation of polymer E is also for its excellent performance within a wide range of concentrations, which is important for a less demanding maintenance of the fluid for companies and in daily life. The next research will be devoted to the endurance and process life of the polymers (especially to E and H) in the forming operation.

## Figures and Tables

**Figure 1 polymers-14-01084-f001:**
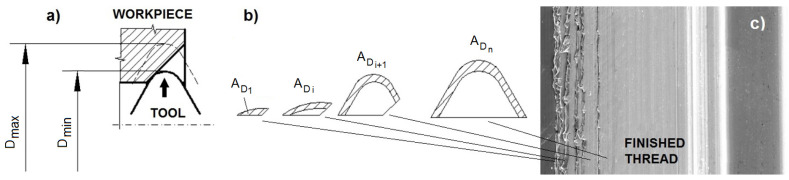
A scheme of individual steps when forming thread: (**a**) the position of the forming tool, (**b**) ascending cross-section of the area to be formed into the thread, (**c**) the surface of the produced thread with typical profile marks at the periphery of the surface.

**Figure 2 polymers-14-01084-f002:**
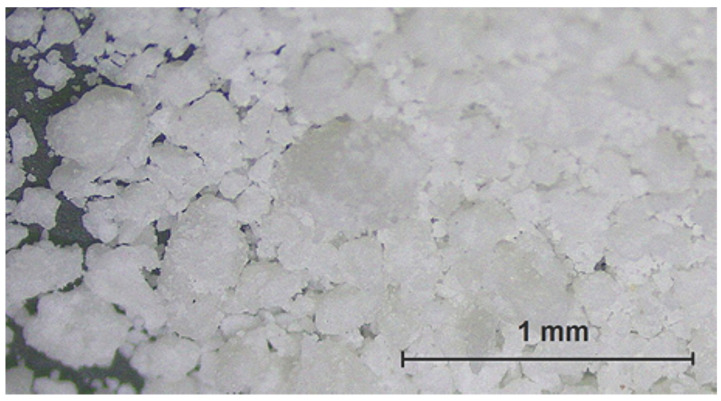
Granules of the polymer soap A.

**Figure 3 polymers-14-01084-f003:**
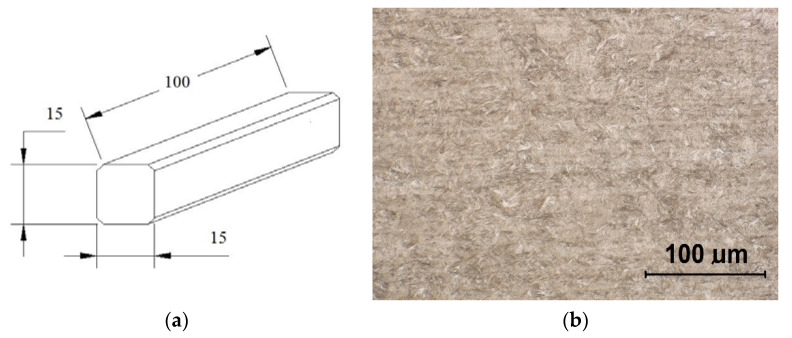
Tested material samples: (**a**) very fine martensitic material structure; (**b**) etched with Nital 2%.

**Figure 4 polymers-14-01084-f004:**
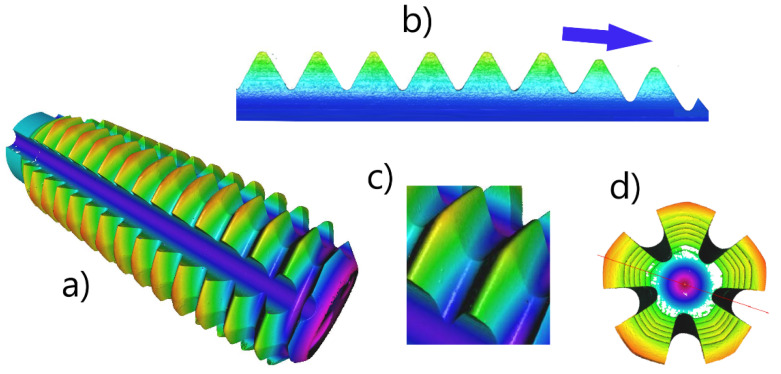
The forming tap geometry—a very good access of fluids to penetrate into tool–workpiece interface; (**a**) an overview of the tool, (**b**) the descending order of the forming tips, (**c**) the wedged profiles, (**d**) a front view of the tool.

**Figure 5 polymers-14-01084-f005:**
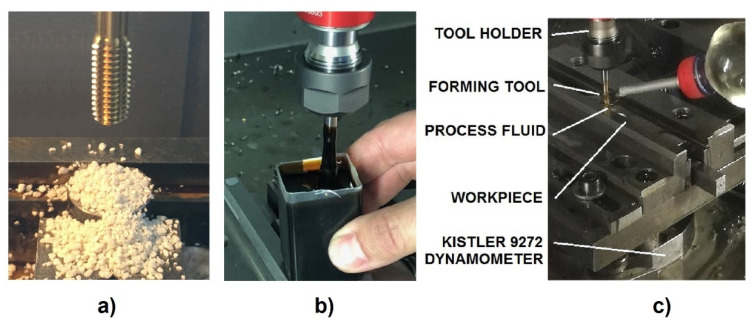
Application of polymers: the soap (**a**), the concentrate (**b**), and the diluted solution (**c**).

**Figure 6 polymers-14-01084-f006:**
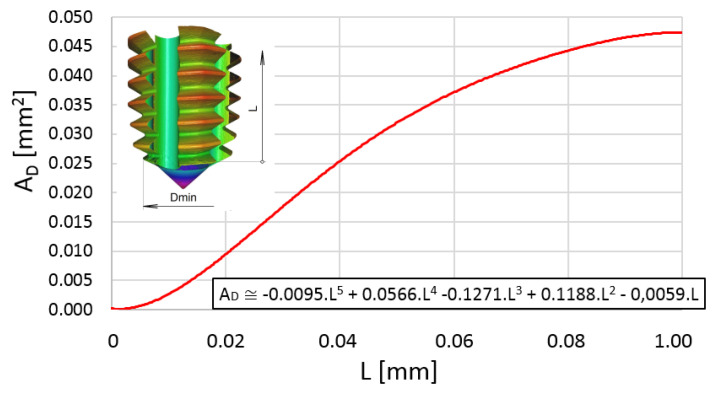
Development of the cross-section (area) of material to be converted into the thread.

**Figure 7 polymers-14-01084-f007:**
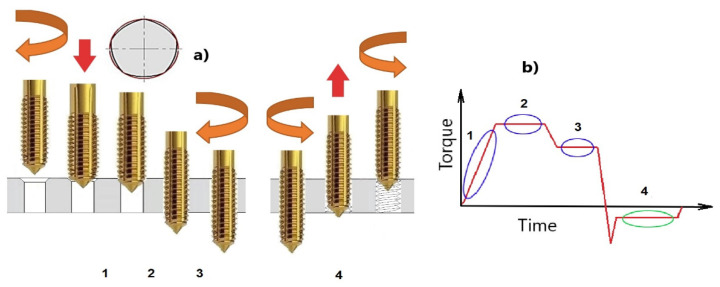
Evolution of the thread production; (**a**) significant positions, (**b**) significant phases for assessment.

**Figure 8 polymers-14-01084-f008:**
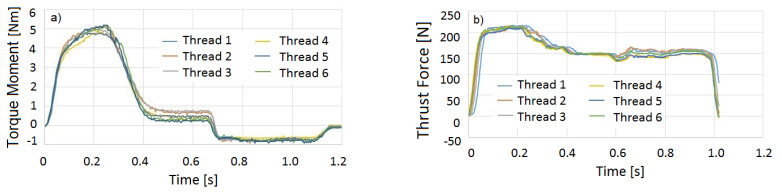
Typical time series measured when using the polymers. (**a**) Torque moment, (**b**) translational force.

**Figure 9 polymers-14-01084-f009:**
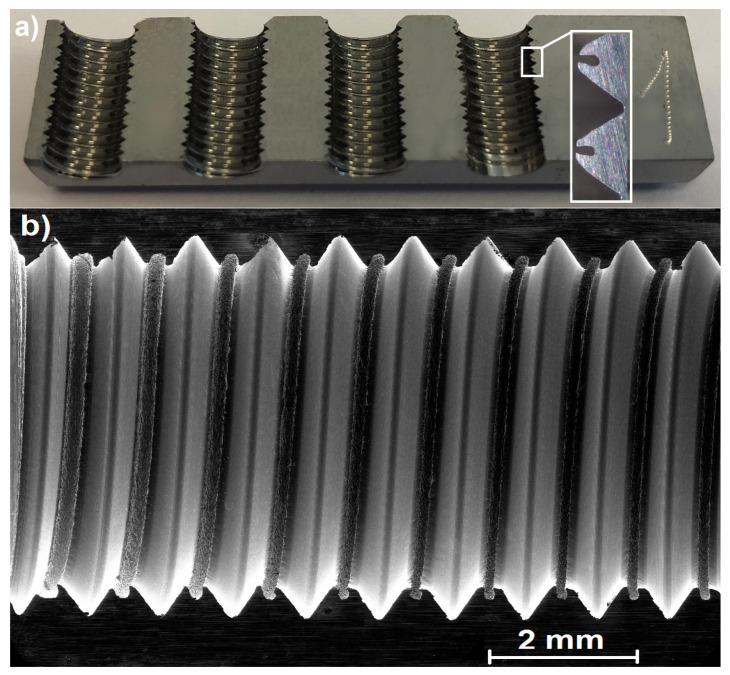
Cross-sections of the threads (**a**) after cutting and polishing; (**b**) the SEM of the cross-section.

**Figure 10 polymers-14-01084-f010:**
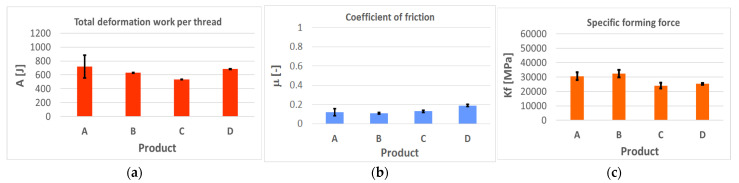
Deformation works, coefficient of friction, and specific forming force for the polymers (**a**–**c**), 100% concentration.

**Figure 11 polymers-14-01084-f011:**
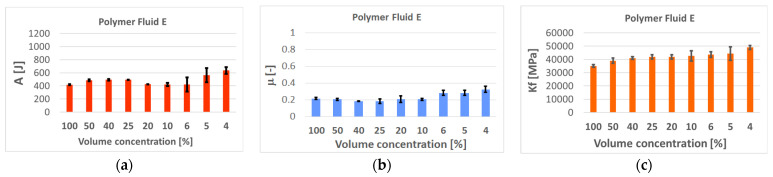
Deformation works (**a**), coefficient of friction (**b**), and specific forming force (**c**) for the polymer **E** and various concentrations.

**Figure 12 polymers-14-01084-f012:**
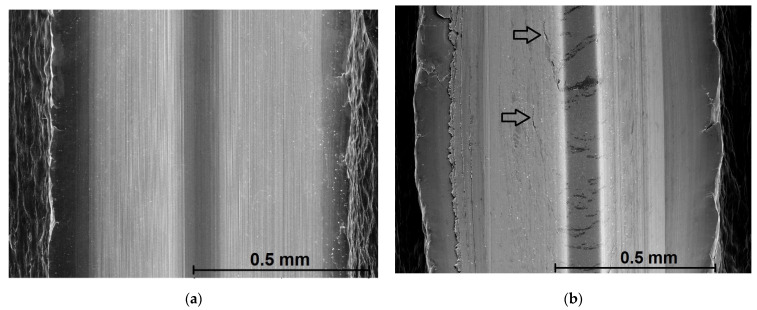
Surface quality of the threads: (**a**) smooth and glossy surface, a few point defects, (Ra 0.6–0.8 μm). This quality was found solely for the lowest coefficients of friction (~0.1). (**b**) Line defects. This quality was found for the coefficients of friction in the range of 0.2–0.3 approximately.

**Figure 13 polymers-14-01084-f013:**
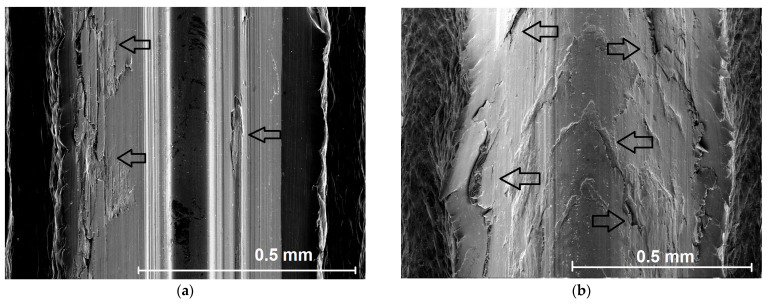
Surface quality of the threads: (**a**) planar defects—for the coefficients of friction around 0.3–0.4, (**b**) bulk defects, abortion of the thread profile (for coefficients of friction equal to or higher than 0.5).

**Figure 14 polymers-14-01084-f014:**
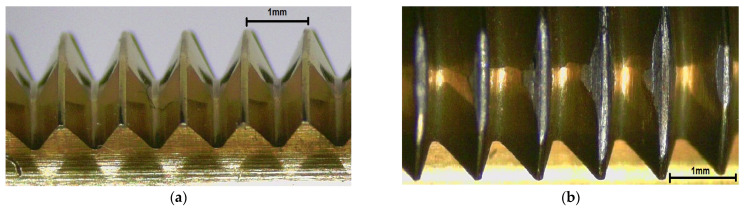
Wear of the taps: (**a**) a new tool, (**b**) at the end of tests.

**Figure 15 polymers-14-01084-f015:**
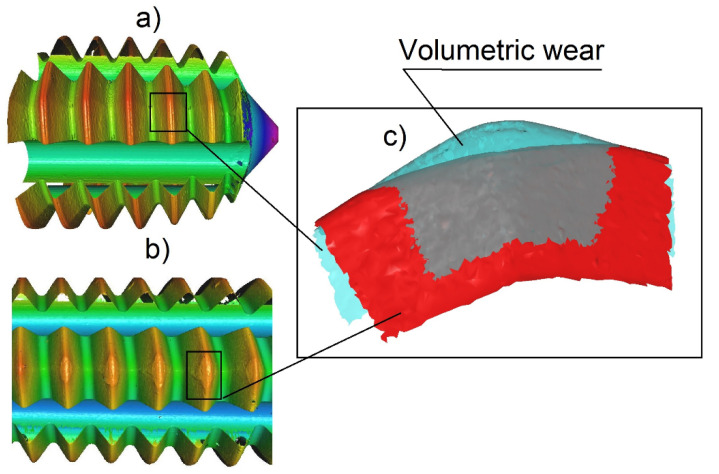
A modern method of wear analysis—measurement of the volumetric wear; (**a**) new tool, (**b**) worn tool, (**c**) a digital subtraction of the surfaces – the worn away material.

**Figure 16 polymers-14-01084-f016:**
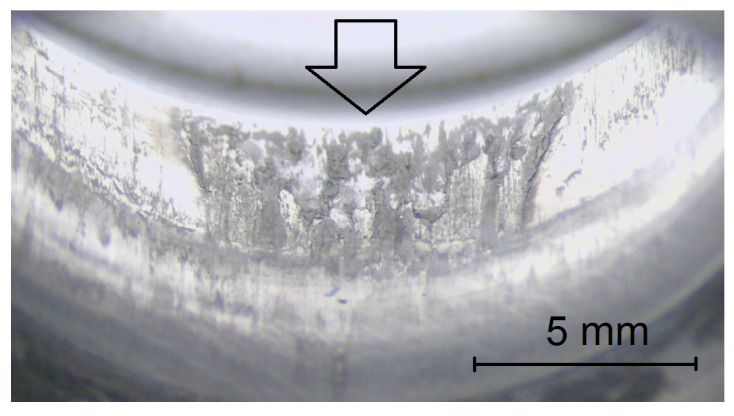
Typical wear of a forming die—irregular wear of the tool due to a poor lubrication with a soap. A higher abrasive loading at the bottom part of the tool (drawing of steel wire in its annealed (“soft”) state) can be seen.

**Table 1 polymers-14-01084-t001:** Chemical composition of the steel according to the standard CSN EN 10089-03.

Chemical	Weight Percent [%]
C	0.51–0.59
Si	1.20–1.60
Mn	0.50–0.80
V	0.10–0.20
P	max 0.025
S	max 0.025
Cr	0.50–0.80
Fe	balance

**Table 2 polymers-14-01084-t002:** Mechanical properties of the steel according to the standard CSN EN 10089-03.

Parameter	Value
Tensile strength	1650–1950 MPa
0.2% proof strength	1600 MPa
Min. elongation at fracture	5%
Reduction in cross-section at fracture	35%
Vickers hardness	248 HV

**Table 3 polymers-14-01084-t003:** Chemical properties of water.

Parameter	Value
Conductivity (mS/m)	27.00
Acidity (pH)	7.72
Total dissolved solids (mg/L)	237
Water hardness (mmol/L)	1.90
Water hardness (°DH)	11.00
Ferrum (mg/L)	<0.10

**Table 4 polymers-14-01084-t004:** An overview of the measured parameters for the tested polymers **A**–**D**.

Product	Volumetric Concentration	Deformation Work [J]	Coefficient of Friction [-]	Specific Forming Force [MPa]
	[%]	mean ± st. dev.	Mean ± st. dev.	Mean ± st. dev.
**A**	100	719 ± 164	0.120 ± 0.035	30,573 ± 2642
**B**	100	629 ± 6	0.110 ± 0.010	32,333 ± 2517
**C**	100	534 ± 1	0.130 ± 0.010	24,000 ± 2000
**D**	100	683 ± 6	0.190	25,275 ± 634

**Table 5 polymers-14-01084-t005:** An overview of the measured parameters for the tested polymer **E**.

Product	Volumetric Concentration	Deformation Work [J]	Coefficient of Friction [-]	Specific Forming Force [Mpa]
	[%]	mean ± st. dev.	Mean ± st. dev.	Mean ± st. dev.
**E**	100	418 ± 8	0.217 ± 0.012	35,067 ± 1007
**E**	50	486 ± 14	0.207 ± 0.012	38,974 ± 2001
**E**	40	493 ± 13	0.183 ± 0.006	41,040 ± 1002
**E**	25	493 ± 4	0.183 ± 0.029	41,833 ± 1607
**E**	20	425 ± 4	0.207 ± 0.040	41,833 ± 1607
**E**	10	420 ± 25	0.207 ± 0.012	42,667 ± 3786
**E**	6	420 ± 107	0.283 ± 0.029	43,667 ± 2082
**E**	5	563 ± 106	0.283 ± 0.029	44,333 ± 5132
**E**	4	636 ± 50	0.325 ± 0.035	49,000 ± 1414

**Table 6 polymers-14-01084-t006:** An overview of the measured parameters for the tested polymer **F**, **H**, and **I**.

Product	Volumetric Concentration	Deformation Work [J]	Coefficient of Friction [-]	Specific Forming Force [MPa]
	[%]	mean ± st. dev.	mean ± st. dev.	mean ± st. dev.
**F**	30	366 ± 20	0.139± 0.026	34,333 ± 1528
**F**	20	667 ± 4	0.137 ± 0.015	30,667 ± 1155
**F**	10	666 ± 6	0.120 ± 0.010	32,433 ± 404
**H**	100	606 ± 49	0.253 ± 0.012	34,333 ± 2082
**H**	40	521 ± 14	0.322 ± 0.031	34,527 ± 410
**H**	20	422 ± 4	0.350 ± 0.036	27,667 ± 2517
**H**	10	292 ± 15	0.270 ± 0.020	26,000 ± 1000
**H**	5	485 ± 44	0.180 ± 0.010	34,167 ± 2137
**H**	1	457 ± 35	0.370 ± 0.061	36,250 ± 1258
**H**	0.1	519 ± 14	0.383 ± 0.025	37,667 ± 1528
**I**	100	934 ± 55	0.530 ± 0.070	55,150 ± 132
**I**	50	632 ± 38	0.313 ± 0.033	51,667 ± 1528
**I**	40	621 ± 26	0.287 ± 0.015	48,333 ± 1528
**I**	25	722 ± 69	0.357 ± 0.040	53,500 ± 1500
**I**	20	894 ± 52	0.573 ± 0.032	56,167 ± 3403

**Table 7 polymers-14-01084-t007:** An overview of the measured parameters for the tested polymer **G**, **G+**, and **G++**.

Product	Volumetric Concentration	Deformation Work [J]	Coefficient of Friction [-]	Specific Forming Force [MPa]
	[%]	mean ± st. dev.	mean ± st. dev.	mean ± st. dev.
**G**	100	606 ± 49	0.300 ± 0.010	32,167 ± 1041
**G**	50	385 ± 8	0.127 ± 0.015	33,833 ± 1756
**G**	30	549 ± 15	0.120 ± 0.014	32,071 ± 1007
**G**	10	648 ± 6	0.143 ± 0.049	34,333 ± 2082
**G+**	50	739 ± 17	0.133 ± 0.021	31,500 ± 866
**G+**	30	755 ± 14	0.145 ± 0.005	27,752 ± 2538
**G+**	10	666 ± 23	0.141 ± 0.010	31,500 ± 866
**G++**	20	517 ± 14	0.343 ± 0.025	32,333 ± 2517
**G++**	10	560 ± 15	0.320 ± 0.017	32,417 ± 2184
**G++**	5	540 ± 14	0.337 ± 0.025	31,500 ± 866
**G++**	3	579 ± 25	0.350 ± 0.036	32,167 ± 1607

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
