# Peer review of "Exquisite Energy Savings at Cold Metal Forming of Threads through the Application of Polymers"

_polymers, 2022, doi:10.3390/polym14061084_

Round 1

Reviewer 1 Report

Miroslav et al. reported the use of different polymers for the energy savings at cold metal forming of threads. This work may be useful for reducing the consumption of energy rises, which qualify the scope of Polymers. However, several problems should be resolved before its publication. A main problem is that there are too many Figures in this long article, and it is hard to grasp your key contribution. Therefore, I strongly recommend to highlight the main findings and results by selecting suitable Figures.

  1. What’s the REM? It should be made clear.
  2. How to evaluate the impact of nano-additives? What’s the underlying mechanism?
  3. The quality of most figures should be improved, such as Figure 9 and Figure 11, etc. In Figure 11, the “a)” and “b)” are lacking.
  4. There are too many Figures in the text. I strongly suggest the authors to combine relevant pictures together, such as Figures 13-20 and Figures 21-22, etc. It is much better to show the Figures associated with your important findings/results in the main text, but some only relevant Figures (in particular for the Experimental) can be provided in the Supporting Information.
  5. Many other problems are also observed, such as “g/l” which should be changed into “g/L”.

Author Response

Ad1) Accepted, it was a misprint, it was corrected.

Ad2)Accepted, it was described in the text more extensively.

Ad3) Accepted, the figures were fixed and improved.

Ad4) Accepted, the text was upgraded and many figures are provided in the Supporting Information.

Ad5) Accepted and the abbreviation was corrected.

Reviewer 2 Report

The authors examined different polymeric materials as lubricants for threading application. The comments and suggestions can be found in the attached documents.

Author Response

All comments and recommendations were accepted and they are reflected in the new upgraded text. The authors would like express to the reviewer sincere thanks for his very professional work.

Reviewer 3 Report

The comments of polymers-1604591 were listed as follows:

(1) It will be great if the authors can provide the information of polymers, at least what they are. While I understand that the limited description due to the commericial interest, the lack of such information will confuse our readers and collegues. 

(2) There are 26 images in the main text, which is too many. It is suggested that the authors either merge some of them or list some of them in the supporting information. 

Thanks. 

Author Response

Ad 1) Yes, it was a tough negotiation with the distributors and companies. They strictly refused to be in a role of "a looser". The descriptions of the polymers come from them and their Polymer Safety Sheets. Nevertheless, the detailed information about the chemicals, agents, concentrations, etc. is many times really missing. So we decided to higlight just two of the products. However, all product belong to well-known producers used in practise today.

Ad) Yes, it was accepted, so we have moved many figures into the Supporting Information. I looks now hopefully shorter and better.

Round 2

Reviewer 1 Report

The authors have made substantial revisions to improve the quality of the manuscript. But there are still too many pictures, and the section of Supporting Information is not good enough.

Author Response

Thank you very much for your careful reading.

All comments have been accepted:

  • several figures have been cancelled,
  • the supporting information was expanded,
  • English was checked again by native English speaker,
  • some other missprints were also corrected.

Reviewer 3 Report

The authors have addressed my concerns. However, it is suggested that the Figures in "Supporting information" would better be listed as "Figure S#" (i..e, Figure S1"). Please revise these and also make the according correction in the main text. 

Thanks. 

Author Response

All comments have been accepted:

  • all documents was revised,
  • some figures cancelled,
  • text corrected,
  • re-numbering of the figures according correction in the main text and the Supporting Information done,
  • all text was checked again by the native English speaker.

Thank you very much for your collaboration and careful reading!